# FedRepre: An Efficient and Scalable Federated Learning Framework with Client Representative Mechanism and Specialized Server Architecture

Yitu Wang*†, Minxue Tang*†, Hanqiu Chen‡, Shiyu Li†, Qilin Zheng†,
Cong Guo†, Andrew Chang§, Cong "Callie" Hao‡, Hai "Helen" Li†, Yiran Chen†

†Duke University. {yitu.wang, minxue.tang, shiyu.li, qilin.zheng, cong.guo, hai.li, yiran.chen}@duke.edu
‡Georgia Institute of Technology. {hanqiu.chen, callie.hao}@gatech.edu
§Samsung Semiconductor, Inc. {andrew.c1}@samsung.com

*Abstract*—**Federated learning (FL) is an emerging distributed machine learning (ML) technique that enables model training across heterogeneous devices while preserving data privacy. However, developing FL in real-world environments faces significant challenges that hinder performance and convergence efficiency. Specifically, the participating devices often have unbalanced local dataset distributions, uneven available computational capabilities, and fluctuating real-time network speeds. Moreover, scaling up the FL system to massive device populations magnifies the importance of the client selection strategy. The execution of such a strategy may emerge as a new bottleneck in the FL system. Unfortunately, prior work has yet to simultaneously address these pressing challenges surrounding real-world FL deployments.**

**We propose FEDREPRE, an efficient and scalable FL framework to accelerate the read-world FL. FEDREPRE introduces a bi-level active client selection strategy called client representative mechanism to guarantee the fast convergence of the global model while reducing the client selection complexity. Specifically, the clients are first clustered based on the statistical correlations, and then cluster selection and representative selection are conducted respectively to attain the maximal global loss decrease and the minimal communication and training latency. To further enhance the scalability, FEDREPRE employs a specialized server architecture to reduce the computation time of the client selection algorithm on the server. We adopt compute express link (CXL) to develop an efficient memory system and unify the memory space with the memory resources on different devices. In addition, we offload the customized hardware selection kernel onto the FPGA with an optimized workflow. We empirically evaluate FEDREPRE across settings with varying scales and heterogeneity levels. The results show that FEDREPRE outperforms previous client selection strategies, achieving $2.16\times - 19.54\times$ speedup of convergence time and up to 1.63% accuracy improvement.**

## I. INTRODUCTION

Machine learning (ML) has permeated various aspects of our daily lives, extending its reach to edge devices such as smartphones, laptops, cameras, and more. The application of ML to diverse user scenarios necessitates the collection and processing of user data from these edge devices, giving rise to significant concerns about user data privacy – an issue we cannot afford to overlook. Federated learning (FL) [8], [9], [16] emerged as a solution to utilize user data from edge devices with a privacy guarantee. FL pushes ML model training to edge devices (also refer to clients) and only aggregates the

*These authors contributed equally to this work.

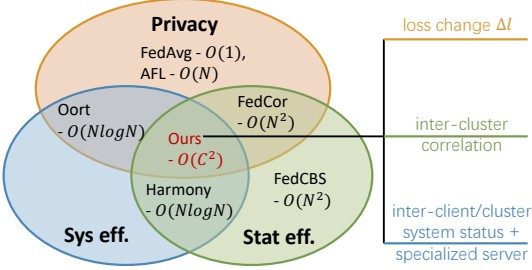

Fig. 1. The comparison of prior federated learning frameworks and ours from the perspectives of privacy, system efficiency, statistical efficiency and scalability (selection complexity).

models from edge devices periodically, thereby avoiding the privacy leakage caused by data transmission.

However, the real-world FL system suffers from poor convergence time-to-accuracy performance (the convergence time to achieve the target accuracy) [7], [14], which is determined by two efficiency metrics, *statistical efficiency and system efficiency*. *Statistical efficiency* is referred to as the number of model aggregation rounds taken to achieve the target accuracy, while *system efficiency* is defined as the duration of each round, including local on-device training latency and data transfer time [26]. Data heterogeneity and system heterogeneity among clients are the major issues that impact the statistical and system efficiency, respectively. Data heterogeneity is manifested in the unbalanced and not independent-and-identically-distributed (non-IID) data among different clients [5]. System heterogeneity is revealed as the variation of runtime computational capability and data transfer speed due to the diversity of devices and networks [12], [24].

Actively selecting a subset of clients to participate in each training round has emerged as a promising technique to tackle the heterogeneity issues [4], [12], [26]. However, previous works on active client selection have deficiencies, as shown in Figure 1. (i) *Compromised **privacy** principle due to disclosed sensitive information*: some previous client selection strategies, FedCBS [32] and Harmony [26], try to achieve an overall balanced data distribution across the selected clients by requiring the clients to report their local data distribution, from which the habits or preferences of the clients can be speculated. (ii) *Low **statistical efficiency** due to uncorrelated*

*client selection*: other client selection strategies like AFL [4] and Oort [12] sidestep privacy-leaking risks by only utilizing the training loss of the local model on each client to guide selection - however, these isolated views neglect the inter-client correlations, thus introducing redundancy and bias into the client selection, which leads to marginal improvement in the statistical efficiency. (iii) *Poor **system efficiency** due to neglected system status and high selection complexity*: a recent study, FedCor [25] proposes a privacy-preserving and statistics-efficient client selection strategy based on the correlations between clients - but the variation in the system status of different clients is neglected, inducing low system efficiency. In addition, the complexity and scalability of the client selection strategies also have a pivotal impact on the system efficiency of an FL system with a massive device population; nevertheless, this fact has been overlooked by previous studies (e.g., FedCor has a quadratic complexity with respect to the number of clients).

As the number of clients increases, we note a shift in the bottleneck of the convergence time for an FL system, *transitioning from client training latency and data transfer overhead to the execution of the client selection strategy on the server*. There are two primary factors that drive the shift. (i) *High selection complexity:* the client selection strategies in previous works have a complexity of $\mathcal{O}(N^2)$ [25], [32] with respect to the number of clients $N$ - this quadratic scaling entails rapid growth in computation and memory demands as N reaches massive populations, creating impractical server-side costs for performing selections. (ii) *Lack of specialized server design:* The inefficiency of executing the selection kernels on the conventional server (just using CPU/GPU to compute) induces high latency, but none of the prior works put forth a dedicated server design for their client selection strategy, thus missing out on the potential to enhance the selection strategy from a hardware perspective. Additionally, the previous works did not explore the memory system design space to optimize the complicated data transmission on the server when dealing with the increasing number of clients.

To tackle the challenges mentioned above, we propose FEDREPRE, an efficient and scalable FL framework with an exquisite algorithm-architecture co-design. **From the algorithm perspective**, FEDREPRE statistically correlates the clients based on their loss changes without requiring any private information. To deal with both statistical and system heterogeneity and reduce the selection complexity simultaneously, FEDREPRE adopts a bi-level client selection strategy called ***client representative mechanism***. The client representative mechanism first clusters the highly correlated clients and selects clusters according to the cluster-level correlation, which can maintain high statistical efficiency while reducing the selection complexity from $\mathcal{O}(N^2)$ to $\mathcal{O}(C^2)$ with the number of clusters $C \ll N$ (e.g., $C = 30, N = 1000$). Then, within each selected cluster, the client representative mechanism picks one client representative with low training time and data transfer latency estimated with the runtime system status, thereby improving the system efficiency.

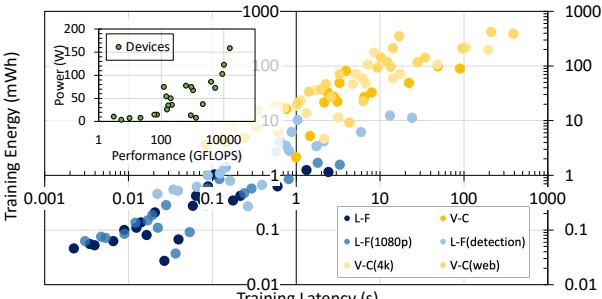

Fig. 2. The heterogeneity of client devices.

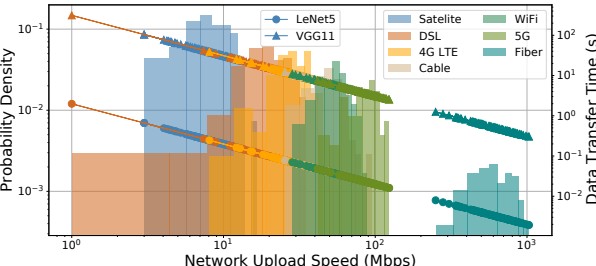

Fig. 3. The heterogeneity of networks.

**From the hardware perspective**, we propose a ***specialized server architecture*** to further reduce the client selection latency induced by the scalability issue. To store a large amount of auxiliary information (embedding vectors) as the number of clients increases, we expand the server memory space to the storage space with compute express link (CXL) [2]. Meanwhile, benefiting from CXL, we design an efficient memory system that further reduces the data transmission latency on the server. To accelerate the execution of the selection kernels, we customize the corresponding selection logic on the FPGA.

A suite of empirical evaluations with different levels of heterogeneity and scale on different workloads reveals the high efficiency of FEDREPRE. when performing the training for 1000 communication rounds, FEDREPRE can averagely accelerate the convergence of the real-world FL system by $2.16\times - 19.54\times$ and reduce the energy consumption by $42\% - 93\%$ compared to the state-of-the-art client selection strategies. Meanwhile, FEDREPRE can improve the model accuracy by up to $1.63\%$. Overall, we make the following key contributions:

**(1)** We propose a novel FL framework FEDREPRE with a low-complexity client selection strategy named client representative mechanism to improve both statistical and system efficiency of the real-world FL system, along with higher model accuracy.

**(2)** We meticulously craft a server equipped with an efficient memory system and specialized selection kernels to further improve the scalability of the FL system.

**(3)** We conduct a comprehensive evaluation for FEDREPRE with the emulation of real-world FL systems, implementation of server hardware kernels, and end-to-end simulation on different workloads in heterogeneous settings.

## II. CHARACTERISTIC STUDY AND MOTIVATION

In this section, we elaborate on the characteristic studies of heterogeneity and scalability in the real-world FL system.

## A. Heterogeneity Issue

The data heterogeneity issue has been fully discussed in prior works [17], [21], from an algorithm perspective with the conclusion suggesting that the data heterogeneity makes the clients unequally contribute to the global loss decrease. Although revealing the local dataset distributions is the most straightforward solution to this issue, on-device data privacy is seriously threatened. Hence, we are motivated to *explore the correlations among the clients implicitly to maximize the global loss decrease*.

For a real-world FL system, the system heterogeneity is more critical to the convergence time because the selected client with the poorest runtime computing capability and lowest network data transfer speed determines the system efficiency. Figure 2 illustrates the heterogeneity of 22 types of typical client devices, including Raspberry Pi 4 [22], MacBook with m1 chip [15], and some other devices on the AI benchmark [1]. The embedded figure shows the variation of theoretical performance and typical power of these devices. Correspondingly, there is a huge difference in training latency and energy consumption when adopting these devices to train different workloads with a batch of 10 samples, e.g., training LeNet5 [13] on FMNIST [30] (L-F) and VGG11 [23] on CIFAR10 [10] (V-C), as shown in Figure 2. In addition, our testing results show that the co-running applications, e.g., 1080p/4k video playback, web applications and neural network-based object detection, on the devices also have an impact on the local training performance. Besides the heterogeneity of client devices, the system heterogeneity also includes the heterogeneity of networks that the client uses. In Figure 3, seven types of networks are shown with different upload speed distributions, along with the corresponding data transfer time for different models. It is not difficult to observe an order of magnitude difference in the data transfer time of the parameters of the same model. The device and network heterogeneity motivate us to *integrate the client-wise system status information, e.g., theoretical performance, co-running applications, and the data transfer speed, into the selection strategy of* FEDREPRE.

## B. Scalability Issue

We observe that as the number of clients increases, the bottleneck of the convergence time shifts to the execution time on the server as shown in Figure 4 (a). We conduct a study on applying FedCor [25] to the training of LeNet5 on FMNIST, selecting 10 clients from 100 and 1000 clients, respectively. The server profiling is performed on two hardware platforms, Intel Xeon Gold 6254 CPU and NVIDIA Titan RTX GPU. We found that when the number of clients increases from 100 to 1000, the server execution takes the most overhead in terms of the end-to-end convergence time, from 7% to 48% on CPU and from 12% to 54% on GPU. Firstly, this is because the selection complexity of FedCor is high, $\mathcal{O}(N^2)$, and quadratically increases with the number of clients. Besides FedCor, in Figure 1, we can observe that all of the prior works trying to improve the statistical or system efficiency

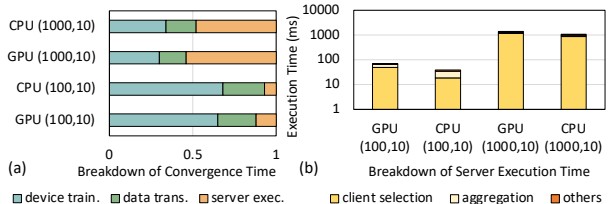

Fig. 4. The profiling results of FedCor [25] with different number of clients. (a) The end-to-end convergence time overhead breakdown and (b) server execution breakdown.

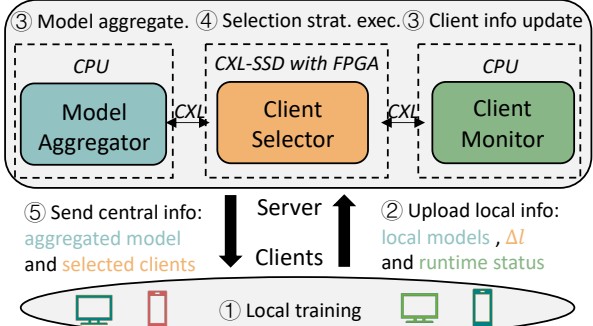

Fig. 5. The overview of FEDREPRE system with 5 stages.

induce high selection complexity, $\mathcal{O}(N \log N)$ or $\mathcal{O}(N^2)$. Thus, we are motivated to *reduce the selection complexity from an algorithm perspective in* FEDREPRE. In fact, the other selection strategies like Oort [12] and Harmony [26] also have the similar issue to FedCor.

Secondly, Figure 4(b) further illustrates the server execution time breakdown. The client selection kernel dominates the server execution time regardless of the client population size. To be specific, client selection kernel takes 46.9% and 79.8% of server execution time when selecting 10 clients from 100 and 1000 clients, respectively. This is because CPU cannot satisfy the high computational parallelism which is required by the client selection logic. In addition, when the number of clients increases, CPU cache cannot buffer all the data and CPU needs to access the corresponding data of clients from DRAM, which makes the client selection kernel takes more overhead. Surprisingly, GPU performs worse than CPU though GPU has more computational power. For example, using GPU takes 300 ms more than CPU when executing the client selection kernel to select 10 clients from 1000 clients. This is mainly because the extra overhead of data copy from CPU to GPU, which is induced by the execution of the PCIe driver [19]. Meanwhile, we also observe that the GPU utilization is only 12% when running the client selection kernel. These interesting facts motivate us to *specialize the server, especially the efficient memory system and customized hardware selection logic, to accelerate the server execution*.

## III. SYSTEM OVERVIEW

Figure 5 illustrates the system overview of FEDREPRE. The whole system consists of a specialized server, which includes a model aggregator, a client selector and a client monitor. The model aggregator and client monitor run on the CPU while the client selector is offloaded and customized

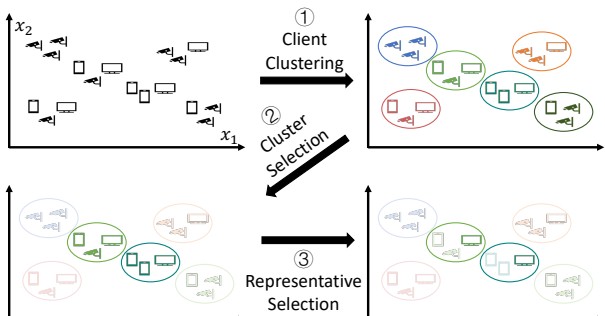

Fig. 6. The workflow of FEDREPRE client selection strategy with client representative mechanism.

on the FPGA within the CXL-SSD. An efficient memory system is developed with the memory resources on the CPU, FPGA and SSD via CXL. In general, the workflow of one communication round in FEDREPRE system can be partitioned into five dependent stages. ① The selected clients, which were generated in the last communication round, train the local model with private datasets; ② The selected clients upload the trained local model to the server; Meanwhile, the runtime system status and the loss decrease ($\Delta l$) could be sent to the server; ③ The model aggregator performs the aggregation of the received local models; At the same time, the client monitor estimates the systematic cost for each client according to the runtime status and could update the inter-client correlations according to the loss decreases; ④ The client selector selects clients for the next communication round by executing our client selection strategy with the client representative mechanism. ⑤ The server sends back the aggregated model and shake hands with the newly selected clients.

## IV. ALGORITHM DESIGN

We elaborate on the algorithm design of FEDREPRE, aiming at mitigating the impact of heterogeneity issues. We first propose the ***client representative mechanism***, which clusters clients according to their statistical correlations and adopts a bi-level selection (cluster selection and representative selection) method to select clients. The client representative mechanism can maintain high statistical efficiency while significantly reducing the selection complexity compared to the previous correlation-based client selection strategy, since only cluster-level correlation is considered in the bi-level selection. Based on the client representative mechanism, we further incorporate the systematic cost into the selection strategy to achieve high system efficiency. Our algorithm is illustrated in Figure 6.

## V. SERVER ARCHITECTURE

We introduce our specialized server architecture to accelerate the client selection procedure. Figure 7 (a) illustrates the overall server architecture design with compute express link (CXL) [2]. The server includes a CPU host with main memory, a GPU, and a CXL-SSD. The CPU host performs the model aggregation and the client clustering. Hence, the corresponding global model, gradients of the selected local models from the clients, and the client-wise system status are stored in the main memory. The CXL-SSD [3] consists of

an FPGA with high-bandwidth memory (HBM) and an SSD. The FPGA is used to implement CXL controller as well as accelerate the inter-cluster selection logic of FEDREPRE. The CPU host communicates with the CXL-SSD through the CXL interconnect. Client embeddings reside in CXL-SSD storage, i.e., SSD, since they are only accessed during occasional re-clustering when GPU updates force new embeddings, which is out of the scope of this work. If all the embeddings are always stored in huge HBM or DRAM, a huge amount of memory resources are wasted while the selection process will not be accelerated. Adopting CXL unifies memory resources across devices, facilitating an efficient hierarchical memory architecture. In such a memory system, the CPU host can access the CXL-SSD using memory load/store instead of initiating the costly I/O access.

## VI. EVALUATION

### A. Experimental Setup and Implementation.

**(i) Datasets and models:** Following previous works [25], [26], [32], we conduct our empirical evaluation on FM-NIST [30] and CIFAR10 [10], with LeNet5 [13] and VGG11 [23], respectively. We train each model for 1000 communication rounds in all our experiments. We partition the training set of each dataset to $N = 100$ or $N = 1000$ clients with two different heterogeneity settings: **(a) DIR:** We use a Dirichlet Distribution with a concentration factor $\alpha = 0.1$ to generate the class ratio for each client. This is a common setting in previous studies [5], [25], [29]; **(b) SC:** We only allocate data from a single class to each client to simulate the extreme data heterogeneity among clients [5], [25].

**(ii) System emulator:** To evaluate FEDREPRE in the real-world FL scenario, we emulate the heterogeneous clients with a device pool consisting of 22 different types of the devices as described in Section II-A, following the methods in the open-source FL benchmark [11]. We randomly sample $N = 100$ or $N = 1000$ client devices from the device pool to simulate FEDREPRE with different scales. Meanwhile, a runtime application pool is built with the 4 tested co-running applications as shown in Figure 2 and a network pool is set up with the 7 network upload speed distributions in Figure 3, where the data is generated from Network Measurements on mobiles [18]. During each communication round, the co-running runtime application and the network situation are sampled from their respective pools to emulate the runtime status and available network bandwidth for each device.

**(iii) Server simulator:** To evaluate the FEDREPRE server, we firstly modify MQSim-CXL [31] by integrating the simulated module of the selection logic to simulate the memory system design with CXL. Furthermore, to get the performance of the specialized selection logic, we implement it on Alveo U50 Data Center Accelerator Card [27] with Vitis HLS [28]. Other server operations are profiled on the real machine, Intel Xeon Gold 6254 CPU [6].

**(iv) Framework implementation:** Integrating the system status from the system emulator and the hardware performance from the server simulator into FEDREPRE framework, the

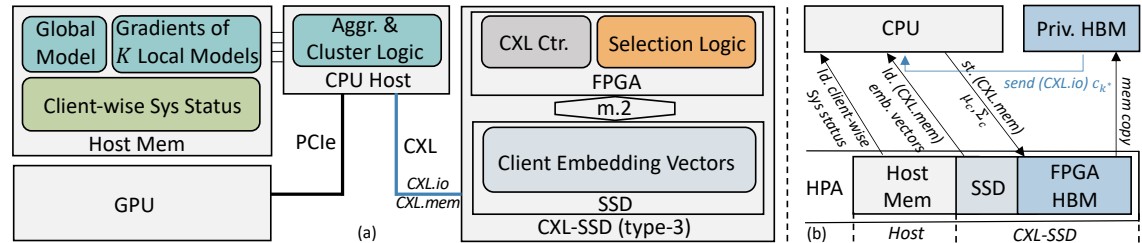

Fig. 7. (a) The server architecture with the corresponding offloaded kernels. (b) The efficient memory system.

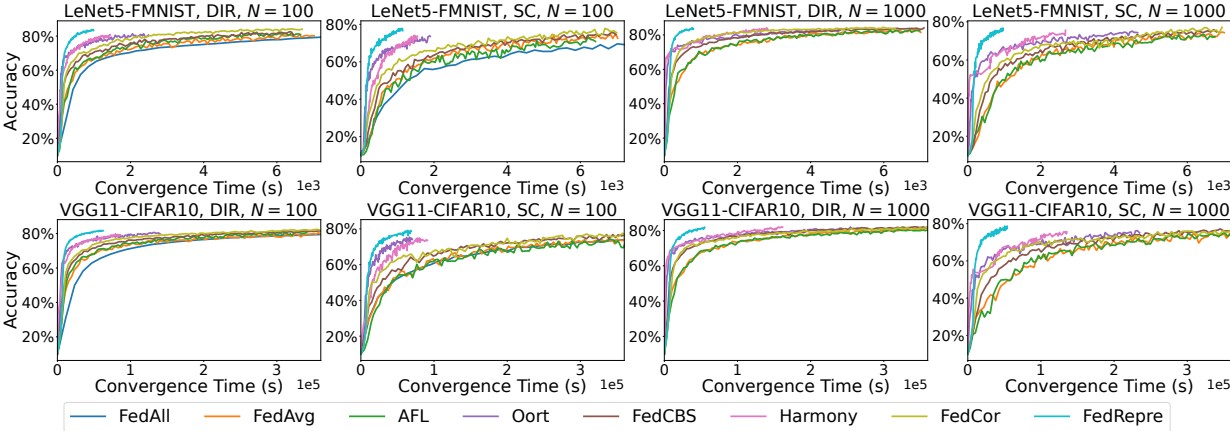

Fig. 8. The convergence time - accuracy curves of adopting different client selection strategies on LeNet5-FMNIST and VGG11-CIFAR10 with 100 and 1000 clients for 1000 communication rounds.

client selection strategy, global model aggregation and client local training are finally implemented in PyTorch [20] to conduct the comprehensive evaluation. For fair comparison, we also implement 7 baselines: FedAll, FedAvg [16], AFL [4], Oort [12], FedCBS [32], Harmony [26] and FedCor [25] with the same system emulator. Except for FedAll, we sample $K = 10$ clients in each round with FEDREPRE and all the other baselines. FedAll is not implemented for $N = 1000$ because it selects all the clients in each communication round, taking too much emulation overhead.

**(v) Evaluation metrics:** We repeat each experiment with three different random seeds and test the global model on the held-out test set. All the results (including test accuracy, number of rounds, amount of time, etc.) are reported as the means over the three random seeds.

*B. Convergence Time - Accuracy*

Figure 8 illustrates the comparison of the convergence time - accuracy curves of various client selection strategies, indicating the overall performance and efficiency of a real-world FL system. In general, FEDREPRE *achieves the shortest convergence time, fastest convergence speed and highest accuracy* on different workloads with both small and large system scales. On average, when conducting the training for 1000 communication rounds, FEDREPRE achieves $19.54\times$, $7.95\times$, $6.61\times$, $6.75\times$ and $6.73\times$ respective speedup over FedAll, FedAvg, AFL, FedCBS and FedCor, mainly because they do not address the system heterogeneity issue when selecting the clients, inducing much longer end-to-end convergence time. While Oort and Harmony consider the system heterogeneity in their selection strategies, Oort suffers from low statistical efficiency without considering the dependency between clients,

and Harmony attains suboptimal system efficiency due to overlooking data transmission time. In contrast, FEDREPRE considers the correlations between clients together with the runtime computational capability and available network bandwidth of each client such that our selection can attain the largest overall loss decrease on all the clients with low training and communication latency. Hence, compared to Harmony and Oort, FEDREPRE can still achieve $2.16\times$ and $3.45\times$ speedups on average, and up to $1.43\%$ and $1.63\%$ accuracy improvements, respectively. When the scale of the FL system increases ($N = 1000$), benefiting from the low selection complexity, FEDREPRE shows even better system scalability and outperforms Harmony and Oort by up to $3.56\times$ and $4.89\times$ speedups respectively. In addition, FEDREPRE guarantees strong privacy protection by only collecting the loss change of each client, while Harmony requires inferring the local data distribution of each client, which definitely compromises the privacy principle in FL.

## VII. CONCLUSIONS

In this work, we propose FEDREPRE, an efficient and scalable FL framework with client representative mechanism and specialized server architecture . The representative mechanism discovers the correlation among the clients, significantly reduces the selection complexity, integrates the client-wise system status, and shows great scalability. Based on our insight that executing the selection strategy could significantly influence the convergence time, a specialized server is proposed, drawing attention to the necessity of a powerful FL server. Based on the experimental findings, FedRepre demonstrates superior performance in terms of convergence time-to-accuracy compared to all state-of-the-art baselines.

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
