# OpenReview forum: "FedRepre: An Efficient and Scalable Federated Learning Framework with Client Representative Mechanism and Specialized Server Architecture"
_iscaconf.org/ISCA/2024/Workshop/MLArchSys — MLArchSys 2024 OralPoster_

### Official Review · Reviewer_YhZa · 2024-05-27
**FedRepre**

**Confidence:** 4
**Rating:** 7

**Detailed Feedback And Questions For Authors:**

Overall, the paper is well written and clear. The method presented in FedRepre provides a novel approach using a bi-level client selection strategy and an CXL memory system to solve issues around statistical and system efficiency for FL.

Other comments/questions:
* Fig 2 is quite cluttered and hard to read. Is there a reason the y-axis tick labels are repeated?
* It would be helpful to spell out what readers should take away, if anything, from the differences in 1080p/4k vs. web applications vs. neural network-based object detection on training.
* Section 2 is a bit long since it repeats concepts introduced in the Intro and some of it may potentially be combined with the Intro.
* Some details are lacking around the client representative mechanism. What do the axes in Fig 6 represent? What are the features being used for the statistical correlations and what is the clustering algorithm used?
* Can more details be provided on the improvements in data transmission latency? (How does Fig 4a. change with FedRepr?)

**Top Reasons To Accept The Paper:**

FedRepre provides a novel method for statistical and system efficiency of FL systems. FedRepre also seeks to improve client selection in FL systems, pointing out previously overlooked problems with client selection complexity and scalability. The authors provide a strong motivation and clear methodology for FedRepre, and the paper demonstrates significant gains in time to convergence and in energy consumption with FedRepre.

**Top Reasons To Reject The Paper:**

None

---

### Official Review · Reviewer_DRFY · 2024-05-28
**Review for FedRepre: A scalable federated learning framework with specialized hardware for client selection**

**Confidence:** 4
**Rating:** 3

**Detailed Feedback And Questions For Authors:**

1. Figure 4 (b) is used as the main motivation to design specialized hardware. However, it's on the logarithmic scale and it's not clear what is the share of other components such as "aggregation" and "others". The figure needs to be normalized to be convincing.

2. The bottleneck of the client selection algorithm needs to be discussed in the paper to justify using a high bandwidth and low latency interface such as CXL and storage hierarchies with SSDs to speed up the server.

3. The metrics discussed in the introduction, i.e., statistical efficiency and system efficiency, are better to be discussed in the evaluation section as well to show the impact of the proposed ideas. It's also better to separate the impact of specifialized hardware and the scalable client selection algorithm, separately.

**Top Reasons To Accept The Paper:**

* Heterogenity and scalability are very important issues in the federated learning algorithms which is targeted by the paper
* The proposed idea is up to 19x better in performance and up to 1.63% better in accuracy, and does not sacrifice privacy

**Top Reasons To Reject The Paper:**

* The paper claims the client selection is the most time consuming part of the server execution time. Figure 4 (b) shows the breakdown of the server time. However, this figure is on logarithmic scale and the other components ("aggregation" and "others") are shown on the top part of the graph, hence it's not accurate to claim they have lower execution time. The graph needs to be normalized to convey this information.

* One of the main idea of the paper is to design specialized hardware for speeding up the client selection algorithm. Assuming the motivation (figure 4(b) mentioned above) is convincing, still it's not clear why the solution to this problem should be as costly as designing a specialized system, leveraging expensive FPGA, CXL and SSDs. More supporting data is needed to show, for example, if the I/O time being the bottleneck in the execution time.

* The paper claims improving the accuarcy by 1.6% and speeds up the learning by 19x compared to previous approaches. However, this comes with a huge dollar cost which was not mentioned in the paper.

---

### Official Review · Reviewer_chpM · 2024-05-29
**The paper presents FedRepre, that introduces a bi-level active client selection strategy called client representative mechanism to guarantee the fast convergence of the global model while reducing the client selection complexity.**

**Confidence:** 3
**Rating:** 5

**Detailed Feedback And Questions For Authors:**

The paper presents an efficient 5 stage framework to reduce the client selection complexity in FL. It demonstrates across MNIST and CIFAR10 with LeNet and VGG model variants to demonstrate faster convergence. However, the current version misses demonstration with algorithms that makes convergence faster in a provable way or with methods to support heterogeneous devices.

**Top Reasons To Accept The Paper:**

The authors introduces a bi-level active client selection strategy called client representative mechanism to guarantee the fast convergence of the global model while reducing the client selection complexity. Additionally, to enhance the scalability, FEDREPRE employs a specialized server architecture to reduce the computation time of the client selection algorithm on the server. The results show that FEDREPRE outperforms previous client selection strategies, achieving 2.16× – 19.54× speedup of convergence time and up to 1.63% accuracy improvement.

The authors' clear demonstration on the training overhead in selection of the clients (Fig. 4).

**Top Reasons To Reject The Paper:**

All the demonstrations were done with older model variants.

All demonstrations were done with simple datsets.

Implication of advanced algorithms to handle client resource heterogeneity (1-2) has not been discussed

1. Overcoming Resource Constraints in Federated Learning: Large Models Can Be Trained with only Weak Clients, TMLR 2023

2. Heterofl: Computation and communication efficient federated learning for heterogeneous clients, ICLR 2022

---

### Official Review · Reviewer_qUer · 2024-05-30
**FEDREPRE presents a clustering algo for levarging heterogeneity and new architecture for scaleble training.**

**Confidence:** 2
**Rating:** 6

**Detailed Feedback And Questions For Authors:**

Please refer to the above section for weakness.

### Writing and Paper improvements:
- Writing Clarity is above average, but further improvement is needed.
- Fig 2 is very congested, and colors are not distinguishable.

**Top Reasons To Accept The Paper:**

- Paper presents innovations in the clustering algorithm and server design for federated training, which provides reasonable accuracy and performance gains.
- Able to leverage the heterogeneity of the clusters to reduce the convergence time.
- Server architecture using CXL-based memory can be a scalable and cheaper solution.

**Top Reasons To Reject The Paper:**

- The algorithm in the paper doesn't indicate why it can have better accuracy compared to other baselines.
       For eg., there might be some missing components in the baseline implementations.
- The algorithm design section of FEDREPRE seems very trivial (lacks details).
- Earlier epochs of FEDREPRE perform worse compared to baselines. For, in N=1000 cases, harmony achieves better accuracy in the earlier epoch; why is this the case?

---

### Decision · Program_Chairs · 2024-05-30

**Decision:**

Accept (Oral/Poster)

**Comment:**

Congratulations! We are pleased to inform you that your paper has been accepted for presentation at MLArchSys 2024. We look forward to your participation at the workshop. Further details regarding the schedule and format will be provided soon. See you at the workshop!